# Increased photosynthesis during spring drought in energy-limited ecosystems

David L. Miller ®[1] ✉, Sebastian Wolf ®[2] ✉, Joshua B. Fisher ®[3], Benjamin F. Zaitchik ®[4], Jingfeng Xiao ®[5] & Trevor F. Keenan ®[1,6] ✉

Drought is often thought to reduce ecosystem photosynthesis. However, theory suggests there is potential for increased photosynthesis during meteorological drought, especially in energy-limited ecosystems. Here, we examine the response of photosynthesis (gross primary productivity, GPP) to meteorological drought across the water-energy limitation spectrum. We find a consistent increase in eddy covariance GPP during spring drought in energy-limited ecosystems (83% of the energy-limited sites). Half of spring GPP sensitivity to precipitation was predicted solely from the wetness index ($R^2 = 0.47$, $p < 0.001$), with weaker relationships in summer and fall. Our results suggest GPP increases during spring drought for 55% of vegetated Northern Hemisphere lands ( >30° N). We then compare these results to terrestrial biosphere model outputs and remote sensing products. In contrast to trends detected in eddy covariance data, model mean GPP always declined under spring precipitation deficits after controlling for air temperature and light availability. While remote sensing products captured the observed negative spring GPP sensitivity in energy-limited ecosystems, terrestrial biosphere models proved insufficiently sensitive to spring precipitation deficits.

Carbon cycling in terrestrial ecosystems can be strongly affected by drought. Drought often limits ecosystem photosynthesis (i.e., gross primary productivity, GPP), reducing the ability of ecosystems to take up atmospheric carbon dioxide ($CO_2$)[1–4]. With the duration and severity of droughts expected to increase with climate change[5], it is essential to understand GPP responses to drought for projecting future changes in the terrestrial carbon cycle[6–9]. This has led to an extensive body of literature focused on understanding the negative impacts of droughts on various aspects of ecosystem function, from photosynthesis[10] to mortality[11].

Many environmental factors (e.g., soil parent material, nitrogen availability, carbon dioxide[12]) can affect GPP, but carbon uptake is primarily controlled by a balance of water vs. energy limitations (e.g., radiation, temperature)[1,13–15]. The strength and direction of these

controls varies across ecosystems, with water availability being the primary driver of interannual variability in GPP at local ecosystem scales[16]. Plants increase GPP by keeping stomata open and losing water to the atmosphere through transpiration in exchange for carbon[12]. If soil water becomes limiting or atmospheric demand through vapor pressure deficit (VPD) becomes too high, vegetation can respond by limiting stomatal conductance and reducing carbon assimilation[11,17–20].

Droughts resulting from low soil moisture often reduce vegetation water availability[3]. Meteorological droughts, in which there is a precipitation deficit, also often correspond with higher temperatures and increased solar radiation with reduced cloud cover[21]. In dry climates, ecosystem productivity is water-limited, having a tight positive relationship with precipitation inputs; this relationship is becoming increasingly sensitive to precipitation with rising

[1]Department of Environmental Science, Policy, and Management, University of California, Berkeley, CA 94720, USA. [2]Department of Environmental Systems Science, ETH Zurich, 8092 Zurich, Switzerland. [3]Schmid College of Science and Technology, Chapman University, Orange, CA 92866, USA. [4]Department of Earth and Planetary Sciences, The Johns Hopkins University, Baltimore, MD 21218, USA. [5]Earth Systems Research Center, Institute for the Study of Earth, Oceans, and Space, University of New Hampshire, Durham, NH 03824, USA. [6]Climate and Ecosystem Sciences Division, Lawrence Berkeley National Laboratory, Berkeley, CA 94720, USA. ✉e-mail: dlm@berkeley.edu; sewolf@ethz.ch; trevorkeenan@berkeley.edu

atmospheric $CO_2$[3,22-24]. In energy-limited ecosystems, GPP may increase even as soil moisture declines, due to higher temperatures and more light availability[25]. This energy-limited response has been shown consistently in tropical rainforests, which are often light-limited[26-29], and in high elevation ecosystems[30,31]. Theory suggests that GPP can increase during negative precipitation anomalies (i.e., meteorological droughts[32]) in mid- and high-latitude ecosystems as well, a response that has found some support[33-37]. For example, in regions that are cold or temperate, drought conditions can result in higher solar radiation and air temperatures that can sometimes increase vegetation activity[21,30,37], such as during the 2012 drought in the United States[38] and the 2018 and 2022 droughts in Europe[4,33]. Due to the lack of research examining vegetation responses across large scales, there remains much uncertainty regarding the response of energy-limited ecosystems to meteorological drought based on observational data.

Here, we examine how the water-energy limitation spectrum affects the sensitivity of GPP to meteorological drought across the Northern Hemisphere (>30° N)[39]. We use a newly available compilation of long-term eddy covariance (EC) observations from >60 long-term sites with ≥10 years of data within a given season (spring (61), summer (62), or fall (63)) (Supplemental Fig. S1, Supplemental Table S1), and analyze GPP sensitivities to drought using precipitation, air temperature, and photosynthetically active radiation (PAR) as meteorological descriptors. Sites are categorized as water- or energy-limited based on the wetness index (WI), which describes aridity and is calculated as long-term mean annual precipitation divided by potential evapotranspiration (PET). We then evaluate how well the EC-observed sensitivities are represented in terrestrial biosphere model (TBM) outputs[40,41], specifically using the TRENDY ensemble of models[42-44], and satellite remote sensing GPP products. We address three central research questions: 1) what is the relationship between aridity and the sensitivity of GPP to meteorological drought for ecosystems >30° N; 2) can we consistently predict when and where GPP will increase or decrease during a meteorological drought in different seasons; and 3) compared to EC observations, how well do gridded estimates from

TBMs and satellite remote sensing estimate GPP sensitivity to precipitation? Using EC observations, we find that ecosystem aridity describes nearly half ($R^2 = 0.47$) of the variability in the sensitivity of GPP to precipitation during spring. Our results demonstrate that during spring meteorological droughts (i.e., precipitation deficits), energy-limited ecosystems routinely increase GPP (83% of the energy-limited EC sites), which is not the case in summer or fall, or for water-limited sites. Satellite remote sensing GPP products capture the increase in GPP during spring droughts that we observe using eddy covariance, but GPP outputs from TBMs are comparatively insensitive to precipitation deficits in energy-limited ecosystems.

## Results
### Seasonal GPP sensitivity to drought using eddy covariance observations

We compared spring, summer, and fall observations to determine how the sensitivity of GPP to drought changes through the growing season. To calculate sensitivity, seasonal sums of GPP from EC were linearly regressed against precipitation, requiring a ≥10 years of data per season (see Methods). During spring drought, energy-limited ecosystems consistently increase GPP (83% of sites, $n = 38/46$; Fig. 1, Supplemental Fig. S2). This yields a negative GPP sensitivity to precipitation. By contrast, water-limited ecosystems have a positive GPP sensitivity to precipitation during the spring (80%; $n = 12/15$ sites). We find that nearly half of the variability in the sensitivity of spring GPP to precipitation across sites can be attributed to the WI ($R^2 = 0.47$, $p < 0.001$).

Although the WI accurately describes changes in spring drought sensitivity across the water-energy limitation spectrum, it performs comparatively poorly in summer ($R^2 = 0.16$, $p = 0.002$; excluding IL-Yat) and fall ($R^2 = 0.30$, $p < 0.001$) (Supplemental Fig. S3). In summer and fall, unlike in spring, energy-limited ecosystems do not have consistently negative GPP sensitivities to precipitation. Relatively few energy-limited ecosystems increase GPP with summer meteorological drought (28%, $n = 13/46$), while fall is mixed (54%, $n = 25/46$). Water-limited ecosystems, however, nearly always increase GPP when there is

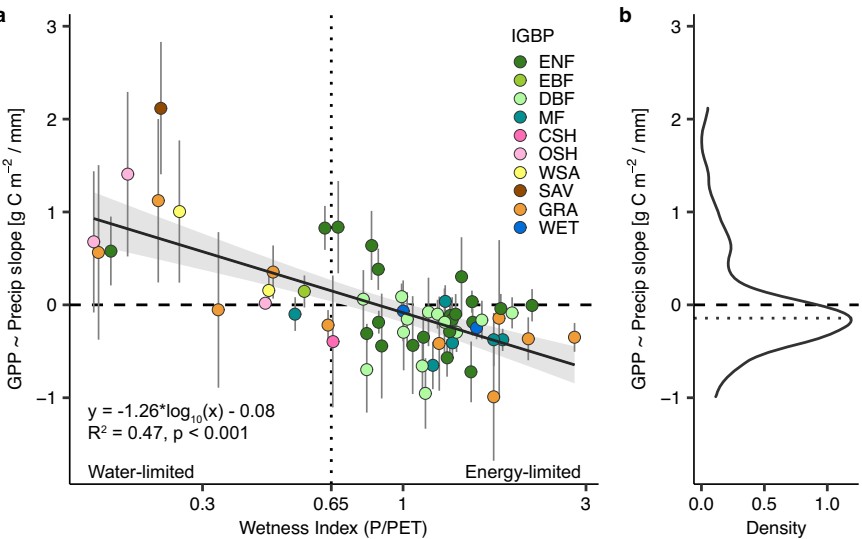

**Fig. 1 | Negative eddy covariance sensitivity of spring GPP to precipitation at high wetness index. a** Simple linear regression of the wetness index (P/PET) to the sensitivity of spring GPP to precipitation ($\log_{10}$-scaled), with each point representing the slope (sensitivity) at a flux tower site with ≥ 10 years of data ($n = 61$ sites, exact $p = -9.001 \times 10^{-10}$). Error bars represent the standard error on each slope. Individual points are colored based on their plant functional type labeled from the International Geosphere-Biosphere Programme (IGBP) classification scheme: ENF

evergreen needleleaf forest, EBF evergreen broadleaf forest, DBF deciduous broadleaf forest, MF mixed forest, CSH closed shrublands, OSH open shrublands, WSA woody savanna, SAV savanna, GRA grassland, WET wetland. Gray shading denotes the 95% confidence interval on the regression. Water-limited sites are to the left of the vertical dotted line at x = 0.65, and the energy-limited sites are to the right. **b** Density plot of the sensitivity of spring GPP to precipitation. The dotted line is the median sensitivity across all sites (median = −0.14).

more precipitation in summer (94%, $n = 15/16$) and often in fall (71%, $n = 12/17$). When we reduce the number of years needed per site to ≥5 per season, nearly doubling the available sites to 112, we find similar patterns in the GPP sensitivities but with lower $R^2$ in spring and fall and no significant trend in summer (Supplemental Fig. S4). We also find similar patterns when evaluating the strength of correlations (Pearson or Spearman) rather than physical units of g C $m^{-2}$ / mm, suggesting that at the ends of the aridity gradient, the GPP sensitivities have not only a greater magnitude but are also more tightly coupled to meteorological conditions (Supplementals Fig. S5, S6). We find similar patterns when substituting the Palmer Drought Severity Index (PDSI) for precipitation but with lower $R^2$ (Supplementals Fig. S7, S8).

We also compare the sensitivity of GPP to light availability (i.e., photosynthetically active radiation; PAR) and air temperature (see Methods). In spring, there are significant relationships between WI and the sensitivity of GPP to PAR ($R^2 = 0.23$, $p < 0.001$), and the sensitivity of GPP to air temperature ($R^2 = 0.27$, $p < 0.001$), but less consistent relationships during summer and fall (Supplemental Fig. S3). Even when controlling for PAR and air temperature in a multiple linear regression in the spring EC observations (see Methods), 61% of energy-limited sites ($n = 28/46$) have a negative GPP sensitivity to precipitation (Supplemental Fig. S9). By contrast, in the partial spring GPP sensitivity to air temperature, 93% ($n = 43/46$) of energy-limited sites have a positive sensitivity. For energy-limited sites, this suggests that warmer springs almost always increase GPP when controlling for precipitation (93%) compared to drier springs when controlling for air temperature (61%). Therefore, our results show that although positive temperature anomalies are the main control on GPP sensitivity during spring for energy-limited sites, meteorological drought is an important and, thus far, overlooked secondary control for ecosystem productivity, particularly as it often co-occurs with warming.

## TBMs and remote sensing differ in strength of spring GPP sensitivities observed from EC

We compared the EC sensitivity results to outputs from an ensemble of terrestrial biosphere models (TBMs; TRENDY v6 S2) and remote sensing products of GPP to determine how well spring GPP sensitivity to precipitation is represented (see Methods). We constructed GPP sensitivity relationships for all models and products (similar to Fig. 1; Supplemental Fig. S10) and then calculated the mean ($\pm$ SE) sensitivities for water-limited and energy-limited sites, respectively. The EC observations have a mean sensitivity for energy-limited sites of $-0.22$ ($\pm 0.05$) g C $m^{-2}$ / mm and for water-limited sites of 0.57 ($\pm 0.17$) g C $m^{-2}$ / mm (Fig. 2). Neither the TRENDY model outputs nor the remote sensing models produce sensitivities similar to EC observations for both energy- and water-limited ecosystems simultaneously. For energy-limited sites, nearly all TBMs underestimate the strength of negative sensitivity of spring GPP to precipitation observed in the EC data (Fig. 2; Supplemental Fig. S10). The TRENDY model mean sensitivity for energy-limited sites is not significantly different from zero at $-0.03$ ($\pm 0.03$) g C $m^{-2}$ / mm ($p = 0.78$, $n = 46$). For water-limited sites, estimates from TBMs spread over a wide range but their mean is not significantly different from the EC observations ($0.54 \pm 0.11$ g C $m^{-2}$ / mm; $p = 0.87$, $n = 15$). By contrast, GPP products driven by remote sensing occupy a different part of the energy-limited vs. water-limited sensitivity space than the TBM outputs. Unlike the TRENDY outputs, remote sensing products produce a reasonable range of sensitivities in energy-limited sites (MODIS-Aqua = $-0.33 \pm 0.05$ to FLUXCOM RS METEO = $-0.02 \pm 0.01$) but underestimate sensitivity in water-limited sites (MODIS-Aqua = $0.11 \pm 0.07$ to FLUXCOM RS = $0.19 \pm 0.06$ g C $m^{-2}$/mm) when compared with EC observations. Raising or lowering the WI threshold for water vs. energy limitations does not change the overall patterns for TRENDY models and remote sensing products compared to EC (Supplemental Fig. S11), and neither

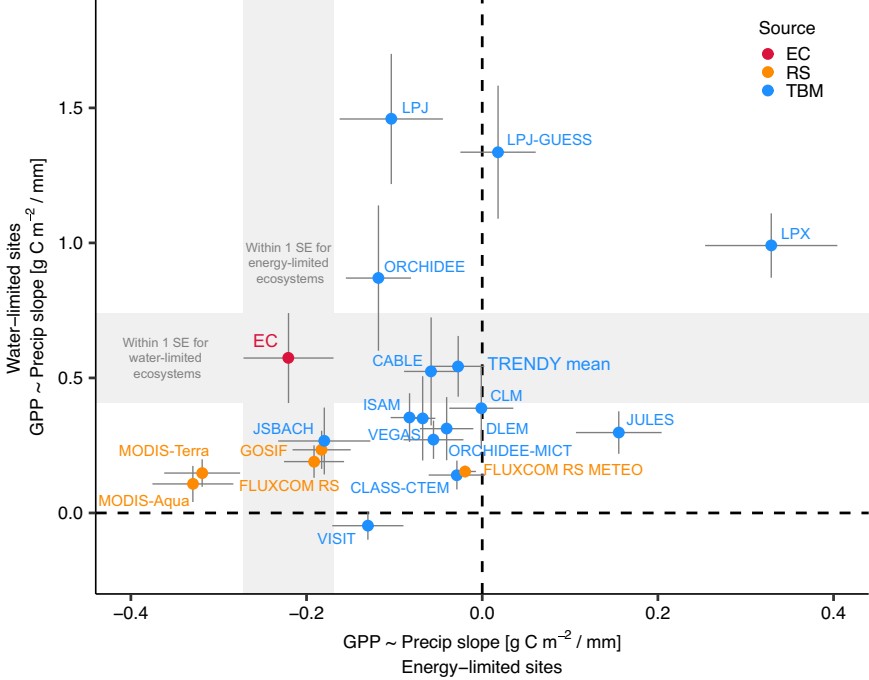

**Fig. 2 | Eddy covariance (EC) observations of spring GPP sensitivities to precipitation occupy a different region of sensitivity space than terrestrial biosphere models (TBM) and satellite remote sensing (RS) products.** Points show energy-limited (x) vs water-limited (y) mean spring GPP sensitivities to precipitation ($\pm$ standard error, SE, bars), with labels for each model or product. Gray rectangles project EC SEs across the plot. Points falling within the gray rectangles have GPP sensitivities within 1 SE of the EC observation means: within vertical rectangle, within 1 SE of energy-limited sensitivities; within horizontal rectangle, within 1 SE of water-limited sensitivities. Due to the different number of sites in energy-limited (46) and water-limited (15) ecosystems, the axes are scaled such that 1 SE is the same size in both directions. References for TBM acronyms are provided in Supplemental Table S2, and RS product descriptions and references are provided in the Methods section 'Remote Sensing GPP Products'. The point 'TRENDY mean' is the mean of all the TBMs.

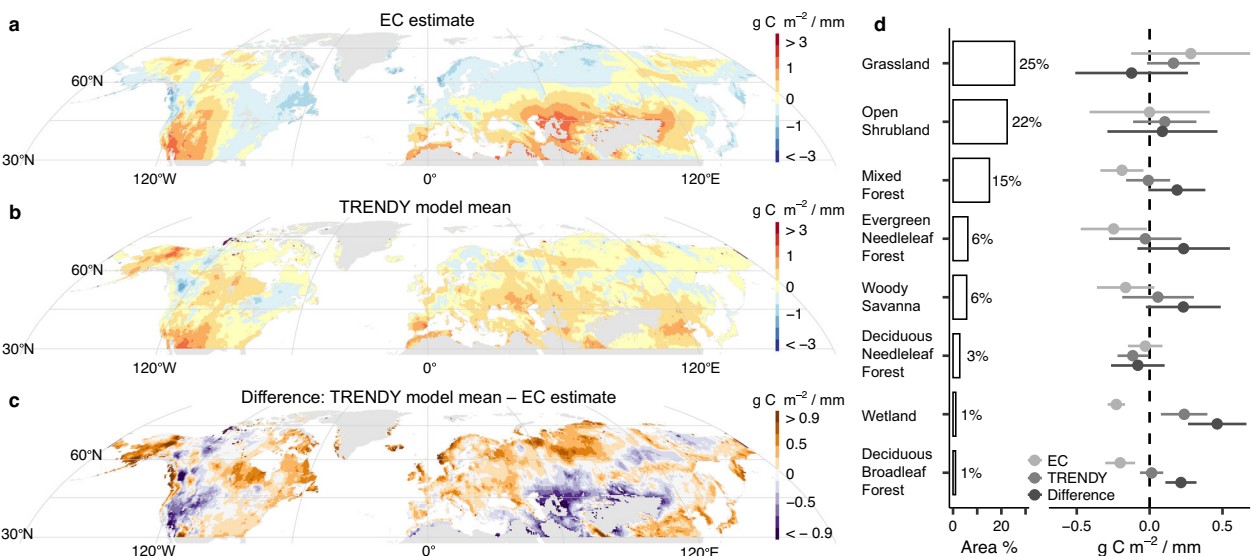

**Fig. 3 | Maps of spring GPP sensitivity to precipitation. a** Slopes derived from eddy covariance (EC) data relationship with wetness index, upscaled using wetness index from TerraClimate climate normals (1981–2010); **b** Mean slopes across all Trends in Net Land-Atmosphere Exchange project (TRENDY) terrestrial biosphere models; **c** Difference between TRENDY model mean slopes and EC estimated slopes such that positive (negative) values indicate that TRENDY is higher (lower) than EC estimate. Base maps of the continents in Fig. 3a–c are in the public domain from Natural Earth. **d** Area (%; bars) of vegetated land from MODIS and mapped mean sensitivities of spring GPP to precipitation (points) from northern land cover types (from IGBP). Areas do not sum to 100%; cover types with areas <1% and urban areas, crops, and crop-natural mosaics are excluded. Error bars on points are standard deviations. Mean and SD are calculated from the middle 95% of data (0.025, 0.975), and are weighted by grid cell area. The number of grid cells by land cover type are: grassland ($n = 6974$), open shrubland ($n = 8298$), mixed forest ($n = 4226$), evergreen needleleaf forest ($n = 1844$), woody savanna ($n = 1919$), deciduous needleleaf forest ($n = 953$), wetland ($n = 379$), and deciduous broadleaf forest ($n = 231$); note that because grid cell size varies with latitude, the grid cell counts are not the same order as the final area (%) in Fig. 3d. Points are shaded to compare the EC-estimated sensitivity (Fig. 3a), the TRENDY model mean sensitivity (Fig. 3b), and their difference (Fig. 3c).

does changing the WI to a larger gridded product (CRU TS 4.05) for the TBMs (Supplemental Fig. S12).

The TRENDY model mean GPP spring sensitivity along the aridity gradient is much more tightly controlled by air temperature ($R^2 = 0.69$) than it is by light availability from PAR ($R^2 = 0.40$) (Supplemental Fig. S13). Both relationships are stronger in the TBMs than they are for the EC observations (air temperature: $R^2 = 0.27$, PAR: $R^2 = 0.23$; Supplemental Fig. S3). The TRENDY model mean has significant relationships within each season for each meteorological variable, except for PAR during summer. This shows more consistent coupling of GPP sensitivity to the meteorological variables in the models than in the EC observations. To test how precipitation, PAR, and air temperature simultaneously affect GPP sensitivity during spring, we performed a multiple linear regression with no interaction (see Methods). Even when accounting for PAR and air temperature, sites in the EC data can have negative spring GPP sensitivities to precipitation, while this is never the case in the TRENDY model ensemble mean (Supplemental Fig. S9). In other words, the models indicate that more precipitation would uniformly correspond to more GPP during spring across the entirety of our aridity gradient, while this is not observed at EC sites.

**Spring GPP sensitivity to precipitation across the Northern Hemisphere**

Our results allow us to map where the model mean of the TBMs is underestimating the strength of the spring GPP sensitivity to precipitation, using the linear regression (from Fig. 1) derived from the EC data and the WI (see Methods). The sensitivity map based on EC has the same spatial patterns as the WI map from which it is derived, with negative slopes in more humid, energy-limited regions and much higher positive slopes in arid, water-limited regions (Fig. 3a). The sensitivity map predicted from the TRENDY model mean, however, does not show a similar spatial pattern, with many more positive than negative slopes, even in humid regions (Fig. 3b). The standard

deviation across the TRENDY models is lowest in humid regions and greatest in arid regions, but there are relatively large differences across models in some regions like north-central Europe (Supplemental Fig. S14). The TRENDY model mean estimates only 36% of the non-barren northern land with a negative spring GPP sensitivity to precipitation, while the EC-WI relationship estimates negative sensitivities at a much larger area of 55%. Among the remote sensing products, patterns of spring sensitivity of GPP to precipitation are not more visually consistent compared to the EC estimate or the TRENDY models (Supplemental Fig. S15).

Subtracting the EC estimate from the TRENDY model mean across space (Fig. 3c), we find that the TRENDY spring precipitation sensitivity of GPP is insufficiently negative and has higher values than EC in many energy-limited regions. These regions include the eastern seaboard of North America; wet mountains, low coastal regions, and the eastern interior of Europe; and a large area of coastal and wet mountainous regions in east Asia. High elevations alone are not necessarily indicative of TRENDY > EC, with TRENDY < EC in large areas of western North America and central Asia that are drier and adjacent to deserts. The TRENDY estimated sensitivity is similar to the EC-derived estimate in many parts of the interior of North America, north-central Asia, and along parts of the Baltic Sea and Mediterranean Sea, in particular regions on the boundary between dry and wet climates, such as the Great Plains of North America. Although there are exceptions in drier parts of northwestern North America and northeastern Asia, higher latitudes often have positive differences.

Comparing these sensitivities based on plant functional types (PFTs) from MODIS land cover, we find that EC sensitivity is far more negative than TRENDY estimates for evergreen needle leaf forests, deciduous broadleaf forests, mixed forests, woody savanna, and wetlands (Fig. 3d). Open shrublands and deciduous needle leaf forests have sensitivities near zero for both EC and TRENDY estimates, perhaps due to cold and/or dry conditions in many of these regions. At the

same time, EC sensitivity exceeds TRENDY estimates for grasslands. Grasslands have the highest variability in GPP sensitivity to precipitation across models (mean SD = 0.29 g C m$^{-2}$/mm) and deciduous needleleaf forests have the least (mean SD = 0.15 g C m$^{-2}$ / mm) (Supplemental Fig. S16).

## Discussion

Several studies have shown examples of how GPP can increase during meteorological droughts in mid- and high-latitude ecosystems[34–36,38]. However, these results have previously been framed as outlier events without broad mechanistic context for how and where these GPP increases are likely to occur. We used long-term EC observations to show that the water-energy limitation spectrum explains the diversity of observed responses of GPP to drought. We show that GPP consistently increases across many energy-limited ecosystems during meteorological droughts in spring, but less consistently in summer or fall, and that the WI alone explains nearly half ($R^2 = 0.47$) of the variability in the direction of spring GPP response to precipitation across sites. This suggests that energy-limited ecosystems often have access to sufficient plant-available soil or ground water storage during meteorological drought in spring, while the energy limitation is removed through co-occurring increases in air temperature and light availability[35,45].

The variability in GPP responses observed in previous studies thus likely reflects the wide variability in site-specific water and energy needs in different parts of the growing season across ecosystems[13,33,45]. While we found water-limited sites nearly always increase GPP with increased precipitation, further seasonal distinctions in GPP responses in energy-limited ecosystems may be related to site-specific factors not captured by the WI[1,35,45–47]. Energy-limited ecosystems that increase GPP with precipitation during the spring have significantly lower PAR ($p = 0.018$) and air temperature ($p = 0.021$) than sites that decrease GPP with precipitation (i.e., increase GPP with meteorological drought; Supplemental Fig. S17). In summer, energy-limited ecosystems that increase GPP with meteorological drought have lower PAR ($p < 0.001$) but are not necessarily colder ($p = 0.22$) than ecosystems that do not (Supplemental Fig. S18). Other energy-limited ecosystems likely are impacted by lagged interseasonal effects of spring droughts, leading to reductions in soil moisture and exacerbated plant stress later in the growing season, particularly in the case of excessive growth and structural overshoot[33,38,39,48]. We must caution that annual GPP can decline overall even with GPP enhancement during spring due to the variety of potential conditions later in the year, and increased GPP does not necessarily equate to increased net carbon uptake due to potential increases in respiration[16,35,38,45].

TBMs are a key tool in predicting the impacts of climatic change on the carbon cycle[40,41,44]. We found that TBMs from the widely used TRENDY ensemble consistently underestimated the strength of spring GPP sensitivity to precipitation deficits in energy-limited ecosystems observed by EC. Despite having much larger grid cells (0.5°) than the flux tower footprints[49], the TBMs had an accurate span of positive GPP sensitivities for water-limited sites, but not negative sensitivities for energy-limited sites. This suggests that GPP-precipitation relationships in many TBMs may be discounting the potential co-benefits of precipitation reductions on spring GPP, and solely focus on precipitation as increasing GPP, which is well-represented for water-limited ecosystems. As a consequence, TBMs may also underestimate the risks of spring drought contributing to later loss of productivity during the growing season, such as through structural overshoot[48,50]. A possible explanation is that TBMs may reduce GPP too strongly under precipitation deficits, which might suggest an excessive GPP sensitivity to precipitation reduction rather than due to co-occurring increases in temperature and solar radiation; for example, in their meta-analysis, Piao et al.[51] found that all 10 of the studied TBMs had a positive GPP sensitivity to variations in precipitation in the Northern Hemisphere

and that TBMs were more sensitive to precipitation than temperature. In addition, the regression of net biome productivity, which is influenced by GPP, to precipitation was overestimated in 9 of 10 of their[51] studied models compared to the observed residual land sink from Friedlingstein et al.[52].

We find a wide array of GPP-precipitation sensitivities across models, with several outliers in how individual models respond to spring drought (Fig. 2; Supplemental Fig. S8). Under conditions of water stress, LPX downregulates photosynthesis to match water demand and supply since it has a supply and demand driven approach[53]. Therefore, photosynthesis and water availability have a positive correlation by design in this model, which may be contributing to the positive correlation between spring GPP and precipitation we observe in Fig. 2 for both energy- and water-limited ecosystems. LPJ, LPJ-GUESS, and ORCHIDEE modify the maximum rate of carboxylation ($V_{cmax}$) to environmental conditions for photosynthetic carbon gain[40,53], and this likely influences the higher sensitivity of GPP to changes in precipitation in these models in water-limited regions. ORCHIDEE-MICT modifies $V_{cmax}$ as well[53], but it does not have the high sensitivities for water-limited sites that are present in ORCHIDEE; this may be due to the higher water holding capacity of soils in ORCHIDEE-MICT[54], such that water-limited sites are more buffered against precipitation deficits. JSBACH in particular does well in comparison to the EC data; this may be due to extensive parameterization for individual PFTs and different stomatal conductance calculations under stressed and unstressed soil water conditions; however, specific attribution is beyond the scope of this analysis and we refer to Reick et al.[55] for more details on this model. Other models that adjust for stomatal conductance (e.g., CABLE, DLEM) also do well for water-limited sites[53], but do not have the sensitivity for energy-limited sites seen in JSBACH.

At the same time, we find remote sensing GPP products span reasonable estimates for GPP sensitivity of energy-limited sites compared with EC, but they underestimate the sensitivity for water-limited sites. This underestimation at water-limited sites is likely related to the lack of explicit parameterization of soil moisture in remote sensing GPP products[10,56], which instead rely on spectral measurements of vegetation structure (e.g., fraction of absorbed PAR), productivity proxies (e.g., solar-induced fluorescence), plant functional type maps, and temperature and VPD[41,57–59]. The expectation is that the soil moisture limitation will be expressed in the vegetation structure or other spectral characteristics (e.g., leaf pigments, water content). Soil moisture (and ground water) are difficult to measure directly across space, but remote sensing-based GPP estimates could likely be improved with explicit characterization of gridded water availability, such as with the Evaporative Stress Index[60] or with newer sensors such as SMAP[61] or GRACE[62].

We upscaled estimates of spring GPP sensitivity to precipitation from EC and the WI to provide a first-order approximation of spatial patterns across the Northern Hemisphere (Fig. 3). These estimates can be directly compared with results from TBMs and remote sensing and can be used as a basis for future comparisons (e.g.[33,38]). However, we acknowledge that, beyond aridity, there are many local environmental factors that vary across different regions and are incorporated by TBMs and remote sensing observations—this includes edaphic or microclimatic factors (e.g., soil texture, water availability, plant structure and diversity)[40,51,56,63]. Within individual TBMs, the diversity of these responses will depend on how variables are parameterized[40]. Our results can be used as a reference to evaluate what would be expected from a climatological perspective (i.e., water vs. energy limitation) alone, and there remains unexplained variability in our regression (Fig. 1).

We find that more than half (55%) of northern non-barren land is estimated to increase GPP with spring meteorological drought, but it is unclear if these spatial patterns will hold with future climate warming.

Warming will enhance primary production during spring for many temperate ecosystems[64]. In humid, mountainous regions where these relationships often occur, climate warming will likely reduce snowpack and subsequent meltwater rates later in spring (e.g., Alps)[46], perhaps limiting GPP during late spring and summer depending on shifts in precipitation patterns[50,63] or continued availability of water in bedrock[65]. Furthermore, changes in these sensitivities may occur at finer regional scales, such as across local elevation gradients: during a warm and dry event, low elevation sites can decrease productivity while high elevation sites may experience an increase due to differences in water availability and energy limitations[30,31]. Changes in GPP sensitivity to precipitation remain uncertain, but potential GPP reductions may be partially offset by rising atmospheric $CO_2$ and possible increases in water use efficiency with warming[66–69].

Our results highlight how GPP can increase in energy-limited ecosystems during spring meteorological drought and reconcile previous results by examining responses through the lens of the water-energy limitation spectrum. Based on EC observations, we show consistent increases in GPP during spring drought in energy-limited ecosystems. The strength of these spring GPP increases is captured in remote sensing GPP products but not by TBMs, highlighting a need for TBMs to better account for the varying effects of meteorological drought on carbon cycling in mid- and high-latitude ecosystems.

## Methods

### Eddy covariance observations

To investigate seasonal differences in GPP sensitivities in temperate ecosystems, we used EC data from long-term, non-cropland sites >30° N. We limited our analyses to the Northern Hemisphere because of our focus on seasonality, land extent, and EC data coverage; there are very few long-term EC sites in the Southern Hemisphere. Croplands were excluded because of possible water inputs from irrigation. Because of our need for long time series of EC data across a wide aridity gradient, we included site data from several sources: FLUXNET2015[70], ONEFlux-Beta (https://ameriflux.lbl.gov/data/download-data-oneflux-beta), ICOS Warm Winter 2020[71], and ICOS Drought 2018[72]. All data were in FLUXNET format at monthly aggregation and processed by the same ONEFlux codebase[70]. If a site had data in both FLUXNET2015 and another more recent release, we used the more recent data set and did not merge or combine across data sources (e.g., we used site US-MMS from ONEFlux-Beta instead of from FLUXNET2015). We developed an initial site list retaining all sites that had ≥5 years of data. The monthly data was aggregated to meteorological seasons (spring: MAM, summer: JJA, fall: SON), taking sums of gap-filled GPP (GPP_NT_VUT_REF). Seasons with >50% of gap-filled GPP data were removed (NEE_VU-T_REF_QC < 0.5, mean from monthly data), as well as any time period that could be documented as having experienced severe disturbance (e.g., fires). Monthly GPP trends were evaluated for all sites to find unflagged discontinuities or errors in GPP, and anomalous years were further investigated prior to inclusion in the analysis. Because longer time series led to more climatically consistent responses, we further limited our sites based on the number of available years, retaining sites that had ≥10 years of data in spring ($n = 61$ sites), summer ($n = 62$ sites), or fall ($n = 63$ sites) (Supplemental Fig. S1, Supplemental Table S1).

At each site and season, we calculated the sensitivity of GPP to meteorological variables present in the FLUXNET format data: gap-filled precipitation (P_F), air temperature (TA_F), and photosynthetically active radiation (PAR = 0.5 * SW_IN_F[73];). We use incident shortwave radiation (SW_IN_F) to estimate PAR instead of photosynthetic photon flux density (PPFD) because SW_IN_F is available across all sites, while relatively fewer sites have PPFD sensors. In the FLUXNET format data, each meteorological variable uses observations at the flux site when available and is filled with data from ERA-Interim when missing[70]. Due to inconsistent availability of in situ meteorological data across sites, we did not distinguish between site-collected and filled meteorological data. To mitigate the effect of plant growth increasing GPP over time, sums of GPP were linearly detrended using Sen's slope if there was a significant trend based on a Mann-Kendall test ($p < 0.05$). Detrending was performed per-season (e.g., there might be an increasing trend in GPP in summer, but not in fall). Meteorological data were not detrended because of the relatively short time series for climatology[74], and differences were minimal when testing detrending precipitation for spring (Supplemental Fig. S19). For each season, we calculated the sum of GPP (g C m$^{-2}$; GPP_sum), sum of precipitation (mm; P_sum), sum of PAR (MJ m$^{-2}$; PAR_sum), and mean air temperature (°C; Ta_mean).

Seasonal GPP sensitivities to meteorological variables were calculated for each site using slopes from linear regressions. Simple linear regressions were used to estimate sensitivity between GPP and meteorological variables, including:

$$GPP_{sum} = \beta_P * P_{sum} + \beta_0 \qquad (1)$$

$$GPP_{sum} = \beta_{PAR} * PAR_{sum} + \beta_0 \qquad (2)$$

$$GPP_{sum} = \beta_{Ta} * Ta_{mean} + \beta_0 \qquad (3)$$

with $\beta_0$ being the intercept for each equation, and the slopes of $\beta_P$ (GPP sensitivity to precipitation), $\beta_{PAR}$ (GPP sensitivity to PAR), and $\beta_{Ta}$ (GPP sensitivity to air temperature).

Correspondingly, multivariate linear relationships were used to assess control by different meteorological conditions. The multivariate relationship predicted GPP based on precipitation, air temperature, and PAR with no interaction:

$$GPP_{sum} = \beta_P * P_{sum} + \beta_{Ta} * Ta_{mean} + \beta_{PAR} * PAR_{sum} + \beta_0 \qquad (4)$$

For all equations, error was estimated as the standard error (SE = SD/n) of the slope terms.

We also compared the trend across the aridity gradient using the sensitivity of GPP to the Palmer Drought Severity Index (PDSI) from TerraClimate at 1/24° (~4 km) for each site[75]. The monthly PDSI value was extracted for each site from the available years for the EC sites that had ≥10 years of data, averaged for each season, and we used the same simple linear regression form:

$$GPP_{sum} = \beta_{PDSI} * PDSI_{mean} + \beta_0 \qquad (5)$$

with $\beta_0$ being the intercept and $\beta_{PDSI}$ being the slope (GPP sensitivity to PDSI). We also tested removing outliers (retained middle 95%) of the sensitivities to determine if anomalous sites had undue influence on the overall trend.

When comparing sensitivities of groups of sites, all statistical comparisons are two-sided student's t-tests.

### Wetness index

For each season and meteorological variable, the GPP sensitivity was regressed against the wetness index (WI). Also known as the aridity index, the WI describes long-term climatological aridity of each site such that greater values are wetter climates. WI was calculated using data for the grid cell containing each site (or nearest grid cell for coastal edges, e.g., IT-Noe) from the TerraClimate climatology product (1981-2010) at 1/24° (~4 km) spatial resolution[75]. WI was calculated as the long-term mean annual precipitation divided by potential evapotranspiration (PET); therefore, lower values are drier and higher values are wetter. In TerraClimate, PET is calculated as reference evapotranspiration ($ET_0$) using the Penman-Monteith equation. We used the threshold definition of 0.65[76], which separates arid and semi-arid from humid ecosystems, to distinguish water-limited (dry; WI < 0.65) from

energy-limited (wet; WI ≥ 0.65) sites. We also tested WI thresholds of 0.5 and 1.0 to evaluate the consistency of responses based on the threshold value. To test if the TBMs were affected by spatial scale, the WI for TBMs was calculated using precipitation and PET from CRU TS v 4.05 (1981-2010) at 0.5° spatial resolution (Supplemental Fig. S12); these changes did not affect the conclusions that the TBMs underestimate the sensitivity for energy-limited sites compared to EC.

## Terrestrial biosphere models TRENDY v6 outputs

We compared EC sensitivities to estimates of GPP sensitivity from TBM outputs using the Trends in Net Land-Atmosphere Exchange project (TRENDY) v6[42–44]. TRENDY is an ensemble of models that estimates ecosystem carbon dynamics from initial forcings. We used outputs from simulation scenario S2, which allows for changes in climate and $CO_2$ concentration through time, but with stable preindustrial land cover (Supplemental Table S2). Outputs are produced at monthly time steps from January 1901 to December 2016. Model outputs initially had different spatial resolutions and were re-gridded to 0.5° prior to analysis[77]. Since all models had the same input meteorological data from CRU NECP v8, for consistency we used precipitation (pr), air temperature (tas), and incident solar downwelling shortwave radiation (rsds, converted to PAR) extracted from the model data provided for the CABLE model at 0.5° spatial resolution[44]. The model SDGVM was excluded because we observed an unrealistic drop in GPP after 2007[78].

The TRENDY model outputs were analyzed for the time period overlapping with the eddy covariance data availability (1992-2016; 1992 is the earliest full year of data from US-Ha1, the Harvard Forest EMS tower site[79]). We did not match individual modeled years with the available years from the observed EC data; we focused on the overall trend in GPP sensitivities across the WI rather than the precise sensitivity at each EC site. GPP sensitivity to precipitation, air temperature, and PAR were calculated per model and season, similar to the EC analysis. To ensure geographic compatibility between the EC data and the models, the model data were extracted for the grid cell which contained each EC flux tower location. We evaluated sensitivities for each model, and averaged GPP across models to develop a TRENDY model mean GPP sensitivity for each flux tower site. The WI for each site was the same as from the EC extracted values (i.e., WI was not regridded to 0.5°).

## Remote sensing GPP products

The EC and TBM results were compared with gridded GPP products derived from satellite remote sensing and meteorological upscaling. Specifically, we used MODIS GPP[59], GOSIF[58], and FLUXCOM[57,80] with time series data extracted for each site location. For each year and season, all GPP products were summed (g C m$^{-2}$ season$^{-1}$). MODIS GPP was extracted at 500 m spatial resolution from AppEEARS from the gap-filled Version 6.0 product at 8-day temporal resolution, for both the Terra (MOD17, 2000-) and Aqua (MYD17, 2002-) satellite platforms. GOSIF (2000-) is derived from OCO-2 SIF soundings that have been spatially and temporally scaled to MODIS surface reflectance time series, from which GPP is calculated. GOSIF GPP was extracted at 0.05° spatial resolution and at monthly temporal resolution. FLUX-COM GPP was estimated from two related products: from remote sensing only (RS V006, 2001-) at 0.083° spatial resolution, and with both remote sensing and meteorological forcing from ERA5 (RS METEO ERA5, 1992-) at 0.5° spatial resolution. Both products were at monthly temporal resolution. All slopes estimated from these products were calculated similarly to the gridded TBMs, with estimated precipitation, air temperature, and PAR data coming from the CABLE model inputs for consistency. Except for FLUXCOM RS METEO ERA5, each time series started near the beginning of the MODIS satellite era (2000-2002) and for this analysis were cut in 2016 at the end of the CABLE meteorological data. We also regridded the remote sensing products (excluding MODIS-Aqua) to 0.5° and estimated spring GPP

sensitivity to precipitation, again using the CABLE precipitation inputs similar to the TBMs.

## Maps of difference in sensitivity between EC, TBMs, and remote sensing

To map the spatial differences between spring GPP sensitivity to precipitation inferred from the EC-WI relationship and from the TRENDY model mean, we calculated WI at 0.5° spatial resolution, spatially aggregating the precipitation and PET from TerraClimate. Next, we used the regression of spring GPP sensitivity to precipitation from EC and the WI (from Fig. 1) to estimate GPP sensitivity to precipitation across the WI grid at 0.5° spatial resolution. The WI-estimated GPP sensitivity was then subtracted from the TRENDY v6 GPP sensitivity to precipitation (difference = TRENDY model mean − estimate from EC observations). We used a 0.5° aggregated land cover class map from MODIS (MCD12 Collection 5)[81] to mask grid cells that were water, barren/sparsely vegetated, or permanent snow/ice. GPP sensitivities within land cover classes were estimated from extents within MODIS land cover, using the middle 95% of the sensitivities from EC and the TRENDY model mean to mitigate the effect of extreme values on the slopes. We calculated the standard deviation of the estimated spring GPP sensitivity to precipitation across TRENDY models and compared these differences across IGBP classes.

## Reporting summary

Further information on research design is available in the Nature Portfolio Reporting Summary linked to this article.

## Data availability

Eddy covariance data are available from: FLUXNET2015: https://fluxnet.org/data/fluxnet2015-dataset/. ICOS Drought 2018: https://doi.org/10.18160/YVR0-4898. ICOS Warm Winter 2020: https://doi.org/10.18160/2G60-ZHAK. ONEFlux-Beta: https://ameriflux.lbl.gov/data/download-data-oneflux-beta/. Other data sources are available from: CRU NCEP: https://rda.ucar.edu/datasets/ds314.3/. CRU TS v 4.05: https://crudata.uea.ac.uk/cru/data/hrg/. FLUXCOM: https://www.fluxcom.org/CF-Download/. GOSIF GPP: https://globalecology.unh.edu/data/GOSIF-GPP.html. MODIS GPP (through NASA AppEEARS): https://appeears.earthdatacloud.nasa.gov/. TerraClimate: https://www.climatologylab.org/terraclimate.html. TRENDY: https://blogs.exeter.ac.uk/trendy/.

## Code availability

Supporting R code available at: https://github.com/dlm4/Drought-GPP-sensitivity. All code used in the analysis is available upon request.

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

## Acknowledgements

D.L.M., B.F.Z., and T.F.K. were supported by an NSF PREEVENTS award (#1854945), and D.L.M. and T.F.K. were supported by a NASA SMAP grant (#80NSSC20K1801). S.W. was supported by a European Commission FP7 Marie Curie International Outgoing Fellowship (#300083) and by ETH Zurich core funding. T.F.K. acknowledges support from a DOE Early Career Research Program award (#DE-SC0021023), a NASA Carbon Cycle Award (#80NSSC21K1705), and from the RUBISCO SFA, which is sponsored by the Regional and Global Model Analysis (RGMA) Program in the Climate and Environmental Sciences Division (CESD) of the Office of Biological and Environmental Research (BER) in the U.S. Department of Energy (DOE) Office of Science. J.B.F. was supported by NASA ECOS-TRESS Science and Applications Team (ESAT) (#80NSSC23K0309) and NASA Earth Science Applications: Water Resources (#80NSSC22K0936). J.X. was supported by NASA's ECOSTRESS Science and Applications Team (#80NSSC20K0167). We thank Maoya Bassiouni, Chi Chen, Kyle Delwiche, Yanghui Kang, Sophie Ruehr, Huiqi Wang and others in the Quantitative Ecosystem Dynamics (Q.E.D.) Group for feedback and data availability, and Hannah Zonnevylle for proofreading and writing style suggestions. Parts of this manuscript were edited for clarity assisted by OpenAI's ChatGPT. Special thanks to the eddy covariance site investigators and the management of FLUXNET, AmeriFlux, and the ICOS community for EC data collection and availability; the TRENDY model teams for model runs and data access; and the MODIS GPP (MOD17 and MYD17), MCD12, FLUXCOM, CRU, and TerraClimate teams for processed data availability.

## Author contributions

D.L.M., S.W., J.B.F., and T.F.K. designed the study. D.L.M. performed the analyses with data and contributions from S.W. and T.F.K. J.X. developed the GOSIF dataset. D.L.M. wrote the manuscript with input from S.W., J.B.F., B.F.Z., J.X., and T.F.K. All authors discussed the results, commented on the figures, and revised the manuscript.

## Competing interests

The authors declare no competing interests.
