## [Peer Review File · Nature Communications]

Increased photosynthesis during spring drought in energy-limited ecosystemsREVIEWER COMMENTS

Reviewer #1 (Remarks to the Author):

The authors investigate the response of GPP to meteorological drought at eddy covariance sites across different seasons. The authors show that energy-limited sites often increase their GPP during spring droughts but not other seasons, with implications for carbon cycling. This is an interesting finding across many EC sites and showing the seasonal aspect of the GPP response is intriguing. As such, the paper is well-suited for publication in Nat Comms and of wide interest.

I do have a few comments about the paper that I hope the authors will consider. If I've understood the methods correctly (L378), the wetness index from EC sites (terraclimate) was used in the TRENDY tower-scale results (Figure 2). At the site scale, there could be quite a large discrepancy between the tower WI and the WI in the forcing dataset that the models "experienced". To confirm this is not a major factor in the obs-model differences, I would suggest that the authors repeat Figure 2 using WI calculated from the model forcings.

I'm also finding the model analysis rather superficial. The authors mainly concentrate on multi-model means and only gloss over the inter-model differences. In several instances, the authors do not even show the model spread (both panels in Figure 3). There is also very little discussion on the potential reasons behind model discrepancies, aside from a few fairly superficial statements. I therefore don't find the model analysis very insightful as the main result of large model differences is hardly surprising. Can you at least provide any explanation for the outlier models (e.g. LPX) or provide more insight based on past literature (one specific suggestion below)?

Specific comments:

L72: clarify that these are long-term mean precip and PET

L84: Would be good to have a couple of sentences here describing how you calculated the GPP-P sensitivity given the journal format of including Methods at the end.

Figure 3: Would be useful to show the model spread spatially to identify areas where models agree most strongly. Similarly it would be useful to get an idea of model spread for the individual biomes

Figure 3: would be useful to have this map repeated for the remote sensing products in the supplementary. Do they show a more coherent spatial pattern compared to EC?

L245: You could check from the data if the energy-limited sites that do not show increased productivity with spring drought are indeed particularly cold/cloudy sites.

L259: There are many papers for individual TRENDY models evaluating simulated GPP and ET against eddy covariance observations during water-stressed conditions. They generally point to land surface/ecosystem models responding too strongly to precipitation deficits (i.e. ET/GPP declining with rainfall, leading to a positive sensitivity) which may be one important factor explaining why the simulated sensitivities in energy-limited sites are generally more positive than observed at EC sites. This may imply a wrong sensitivity to precip declines rather than the models not capturing the co-benefits of precip decline (i.e. higher temp and/or radiation). This should be discussed as one possible explanation, making reference to past studies.

L325: Should this be ERA-Interim?

L328: what about cases where increasing GPP was driven by met conditions rather than plant growth (particularly at mature sites)? In this case you have detrended the GPP but not the corresponding met data. Can this introduce biases in the analysis?

L336: GPPsum is not defined in the text? For clarity, would it be better to use different "GPPsum" names for the P, PAR and Ta equations? E.g. Psum_P, Psum_PAR and Psum_Ta

L350: Do you mean the grid cell containing the site?

L370: please briefly state why SDGVM was excluded rather than just including the reference.

L372: Did you match the years to the EC sites or use 1992-2016 for all TRENDY site pixels? Are your results sensitive to this assumption if you didn't match the years?

Reviewer #2 (Remarks to the Author):

Miller et al. perform an analysis of spring GPP sensitivity to precipitation based on eddy-covariance (EC) data from >60 sites with >10 years observation distributed over Europe and North America and encompassing a variety of non-crop biomes. Precipitation sensitivity is defined as the slope of the linear relationship between spring GPP and spring precipitation. Miller et al. show that half of sensitivity variance across space can be predicted from a climate wetness index ($WI = P/PET$), with arid sites exhibiting a positive sensitivity, while wet sites exhibit a negative one. Miller et al. then compare EC sensitivity to that of a vegetation model ensemble and remote sensing and GPP products and show that models tend to overestimate the precipitation sensitivity of wet sites while remote sensing products underestimate that of arid sites. Finally, the authors use the fitted linear relationship between EC sensitivity and WI to extrapolate the sensitivity over the Northern Hemisphere and spatially characterize the mismatch between EC and model sensitivity. While I found overall the paper to read well and to be interesting as it quantifies a perhaps understudied phenomenon and provides useful benchmarking of vegetation models and remote sensing products, I have several concerns.

My first concern is regarding the definition of (meteorological) drought in the paper. While this is a key concept in the paper and is rightly so argued to matter for C cycling under climate change, drought is nowhere clearly defined in the paper. This is a (minor) issue in itself as meteorological drought can have several definitions and should thus be made clear from the beginning. A more in depth read of the paper suggests that drought is only indirectly defined through GPP sensitivity to precipitations. Thus, when the authors argue that GPP increases during spring drought in energy-limited sites, what they mean is that GPP is negatively correlated to precipitation at these sites. In my opinion this is not quite equivalent. Although this result does suggest that spring precipitation deficit enhances GPP, a more thorough definition of meteorological drought taking into account temperature and atmospheric water demand (e.g., based on meteorological drought indices such as SPEI or PDSI) might have yielded different results. This is even more relevant in the context of climate change, as globally drier conditions may arise essentially as a result of increased temperature and VPD, whereas changes in precipitation pattern will be more heterogeneous. As shown by increasingly positive sensitivity of spring GPP to temperature and radiation across the climate wetness gradient, and as discussed by the authors, the paper main result is most likely driven by the positive effect of temperature and/or radiation at sites where water is not limiting. Taking into account the effect of temperature and radiation leads to much lower sensitivity to precipitation, in absolute value. Hence a more adequate conclusion would be that GPP at energy-limited sites is positively related to temperature, which is a well-known result.

Comparison of EC vs model and remote sensing estimate of GPP sensitivity to precipitation is valuable. However, as is, I don't think that the spatial analysis of model-EC differences (i.e., Figure 3) adds much to the conclusion from Figure 2 that models tend to overestimate GPP sensitivity to precipitation. That is because vegetation models account for much more complexity than the EC extrapolation, which is based only on WI, does. Hence much of the spatial differences may be due to factors ignored by EC extrapolation. For example, from Figure 1, it seems that broadleaf forests tend to have a more negative sensitivity to precipitation than needleleaf, which may reflect contrasting species physiology or forest structure. Ignoring these differences may lead to overestimating the model error over needleleaf forests while underestimating it in the case of broadleaf. Soil factors (water capacity, rooting depth) are likely to be important factors as well of variations in GPP sensitivity to precipitation and driving model-EC differences.

Reviewer #3 (Remarks to the Author):

Summary:

This study integrates GPP datasets from in-situ EC flux towers, remote sensing, and DGVM simulations to investigate the sensitivity of spring GPP responses to meteorological drought. The flux tower data analysis shows that the 'gross' sensitivity of spring (MAM) GPP to inter-annual variability in precipitation, calculated with simple linear regression, is highly correlated with site aridity (wetness index). In moist, more energy-limited sites, the GPP~precipitation sensitivity can be negative, which is mostly driven by associated increases in temperature during such meteorological drought. Comparison with TRENDY model simulations suggests the models overall underestimate the magnitude of such negative precipitation sensitivity. Meanwhile, the remote sensing data set can capture the negative sensitivity but cannot capture the positive sensitivity in water-limited ecosystems. Finally, the study analyzes spatial patterns of model-data differences in spring GPP sensitivity to precipitation by empirically scaling site-level sensitivity to the whole northern hemisphere.

Comments:

The paper is well written. The analysis is comprehensive and careful given so many different data sets are involved. The figures are nice and easy to read.

My main concern is that the lack of unwrapping of 'drought' in the title and main text is misleading and confusing. For example, Fig. 1&2 both focus on the idea that models cannot capture the 'negative' GPP sensitivity to precipitation. However, as the authors recognized in the study, 'meteorological drought' is often accompanied by higher radiation and temperature. In fact, Fig. S7 nicely shows the partial effect of GPP to water, radiation, and temperature. It is interesting to see that the average partial sensitivity of GPP to precipitation in energy-limited systems is not negative but around 0 (Fig. S7 panel) while TRENDY model is positive. This indicates that the models are too sensitive to moisture and underestimate sensitivity to temperature (e.g. phenology). I am worried that the current framing of the manuscript can result in an erroneous interpretation that 'moisture-dependence' is wrong in the models, which is not the case based on Fig. S7. Overall, I suggest using the 'partial sensitivities' instead of gross sensitivities in the main text.

Some minor comments:

Line 29. this is inaccurate, drought does not increase GPP. Associated temperature and radiation increase (mostly temperature based on the analysis) drives GPP increase.

Line. 43-44. To me, energy-limited denotes radiation-limited (e.g. the classic paper by Nemani et al. 2003) in contrast to temperature and moisture limited. Temperature is surely one kind of energy limitation. It is ok for such a generalization but more clear definition would be better here.

Line 91, Fig.1 Why not put 4 rows in this figure? First row is gross sensitivity to precipitation, then followed by partial sensitivity to precipitation, radiation, and temperature. As I expressed above, this unwrapping is important.

Line 106-107, 'Few energy-limited', there is still a quarter of sites, which is more than 'few'.

Line 157, I am curious why some models have closer sensitivity to EC? e.g. JSBACH. Does JSBACH have better moisture dependence and phenology? Include such a figure for partial sensitivity would be helpful as well.

Line 330-331. Why meteorological data was not detrended?

REVIEWER COMMENTS

MAJOR REVISIONS

Reviewer #1 (Remarks to the Author):

The authors investigate the response of GPP to meteorological drought at eddy covariance sites across different seasons. The authors show that energy-limited sites often increase their GPP during spring droughts but not other seasons, with implications for carbon cycling. This is an interesting finding across many EC sites and showing the seasonal aspect of the GPP response is intriguing. As such, the paper is well-suited for publication in Nat Comms and of wide interest.

I do have a few comments about the paper that I hope the authors will consider. If I've understood the methods correctly (L378), the wetness index from EC sites (terraclimate) was used in the TRENDY tower-scale results (Figure 2). At the site scale, there could be quite a large discrepancy between the tower WI and the WI in the forcing dataset that the models "experienced". To confirm this is not a major factor in the obs-model differences, I would suggest that the authors repeat Figure 2 using WI calculated from the model forcings.

Thank you for this comment. We used the wetness index (WI) derived from TerraClimate PET and precipitation to estimate site aridity, and we agree there is potentially a difference between the tower scale WI and model scale WI because of spatial scaling factors. However, we decided to use TerraClimate WI to have a consistent index across all models and remote sensing variables to have a fair comparison.

The reviewer is correct though that the difference in forcing data used could potentially affect the results. In order to test whether this is a major factor in the observation-model difference, we have followed the reviewer's suggestion and reproduced Figure 2 using WI calculated from the model forcings (CRU TS). Specifically, we used data from CRU TS 4.05 for the TBMs at the same time range as the TerraClimate normals (1981-2010) (Supplemental Figure S12). CRU TS 4.05 is at a 0.5 degree spatial resolution (same as regridded TBMs used here), is from a similar base source to the original forcings that were used for the TRENDY models (CRU NECP v8), and is available at monthly time ranges. Although using CRU to calculate WI resulted in some of the individual points for the TBMs shifting by a small amount, the overall conclusions of Figure 2 remain unchanged (i.e., energy-limited sites in TBMs underestimate the negative sensitivity shown in EC). This shows that any discrepancy between tower WI and model WI does not influence the results that we report. We have added this new figure as Supplementary Figure S12 to clarify this point to the readers.

Above: Figure 2. All points are using the TerraClimate WI to determine energy- vs- water limited sites (split at threshold of WI = 0.65).

Supplemental Figure S12: Changing TBM WI input from TerraClimate to CRU TS 4.05 shifts some TBMs a small amount but does not change the overall conclusions from Figure 2: the TBMs underestimate the negative sensitivity from energy-limited sites that is shown in the EC data.

Figure 2 is energy-limited vs. water-limited sites, and so the wetness index itself is not directly on the plot, but rather the grouping of which EC sites are considered energy-limited (x) or water-limited (y). Within this plot, some sites shift their location depending on which side of the WI range they are on, thus affecting the distribution of sites within the TBMs (the sites that are considered energy or water limited), but this does not change the overall structure of Figure 2.

In addition, Supplemental Figure S11 (new numbering) shows what Figure 2 would look like if we changed the WI threshold for energy vs water limited, and it shows the same raise the threshold to

Supplemental Figure S11: Even when changing WI thresholds, EC observations of spring GPP sensitivities to precipitation occupy a different region of sensitivity space than models and satellite products. Same as Figure 2 but changing the WI threshold to (a) 0.50 and (b) 1.00. (The version shown in Figure 2 in main text is listed at 0.65.) All axes are scaled such that 1 SE is the same size in both directions within each plot.

I'm also finding the model analysis rather superficial. The authors mainly concentrate on multi-model means and only gloss over the inter-model differences. In several instances, the authors do not even show the model spread (both panels in Figure 3). There is also very little discussion on the potential reasons behind model discrepancies, aside from a few fairly superficial statements. I therefore don't find the model analysis very insightful as the main result of large model differences is hardly surprising. Can you at least provide any explanation for the outlier models (e.g. LPX) or provide more insight based on past literature (one specific suggestion below)?

Thank you for this recommendation, other reviewers had concerns about this as well. We updated the paper to incorporate this suggestion, in particular we expanded the discussion and interpretation of the between-model differences.

We now added a new text (L279-286, line numbers refer to clean version of manuscript): "A possible explanation is that TBMs may reduce GPP too strongly under precipitation deficits, which might suggest an excessive GPP sensitivity to precipitation reduction rather than due to co-occurring increases in temperature and solar radiation; for example, in their meta-analysis, Piao et al. (2013) found that all 10 of the studied TBMs had a positive GPP sensitivity to variations in precipitation in the northern hemisphere and that TBMs were more sensitive to precipitation than temperature. In addition, the regression of net biome productivity, which is influenced by GPP, to precipitation was overestimated in 9 of 10 of their studied models compared to the observed residual land sink from Friedlingstein et al. (2010)."

Following this, we also added a new paragraph in the Discussion on model specific details explaining differing sensitivities, new L287-305:

"We find a wide array of GPP-precipitation sensitivities across models, with several outliers in how individual models respond to spring drought (Figure 2; Supplemental Figure S8). Under conditions of water stress, LPX downregulates photosynthesis to match water demand and supply since it has a supply and demand driven approach⁴⁶ (Pachalis et al., 2020). Therefore, photosynthesis and water availability have a positive correlation by design in this model, which may be contributing to the positive correlation between spring GPP and precipitation we observe in Figure 2 for both energy- and water-limited ecosystems. LPJ, LPJ-GUESS, and ORCHIDEE modify the maximum rate of carboxylation (V_{cmax}) to environmental conditions for photosynthetic carbon gain^{35,46} (Fisher et al., 2014; Pachalis et al., 2020), and this likely influences the higher sensitivity of GPP to changes in precipitation in these models in water-limited regions. ORCHIDEE-MICT modifies V_{cmax} as well⁴⁶ (Paschalis et al., 2020), but it does not

have the high sensitivities for water-limited sites that are present in ORCHIDEE; this may be due to the higher water holding capacity of soils in ORCHIDEE-MICT⁴⁷ (Guimberteau et al., 2018), such that water-limited sites are more buffered against precipitation deficits. JSBACH in particular does well in comparison to the EC data; this may be due to its many sub-models for individual plant functional types (PFTs) and different stomatal conductance calculations under stressed and unstressed soil water conditions; however, specific attribution is beyond the scope of this analysis and we refer to Reick et al.⁴⁸ (2021) for more details on this model. Other models that adjust for stomatal conductance (e.g., CABLE, DLEM) also do well for water-limited sites⁴⁶ (Paschalis et al., 2020), but do not have the sensitivity for energy-limited sites seen in JSBACH.

We have also added more supplemental figures (Figures S13, S15) to support this further based on specific comments for Figure 3. Figure S14 shows the standard deviation of the GPP~precipitation sensitivity across models, with humid regions having the lowest SD and arid regions having the highest. Figure S16 shows the standard deviation of each IGBP vegetation class, with grassland (GRA) having the greatest variation in estimated sensitivity across models (see below comment on Figure 3).

Supplemental Figure S14: Standard deviation in GPP~precipitation sensitivity across all included TRENDY models during the spring. Models are most consistent (lowest SD) in humid regions and least consistent (highest SD) in arid regions with some exceptions: north-central Europe has relatively high SD despite being humid, and western China has relatively low SD despite being arid.

Supplemental Figure S16: Standard deviation across all included TRENDY models by biome. Grasslands have the greatest variation in estimates across models.

Specific comments:

L72: clarify that these are long-term mean precip and PET

Changed to "...and is calculated as long-term mean annual precipitation divided by potential evapotranspiration (PET)."

L84: Would be good to have a couple of sentences here describing how you calculated the GPP-P sensitivity given the journal format of including Methods at the end.

Added new sentence: "To calculate sensitivity, seasonal sums of GPP from EC were linearly regressed against precipitation, requiring a minimum of 10 years of data per season (see Methods)."

Figure 3: Would be useful to show the model spread spatially to identify areas where models agree most strongly. Similarly it would be useful to get an idea of model spread for the individual biomes

We have added new figures to the supplemental material (S14, S16; see earlier comments for figures) and new text to the Results section "Spring GPP sensitivity to precipitation across the Northern Hemisphere" (~new Lines 196-198). The new figures show model SD spatially and SD within biomes: "The standard deviation across the TRENDY models is lowest in humid regions and greatest in arid regions, but there are large relatively large differences across models in some regions like north-central Europe (Supplemental Figure S14)."

The new supplemental figure showing difference in SD across biomes and new text at lines 222-224: “[Grassland] GRA has the highest variability in GPP sensitivity to precipitation across models (mean SD = 0.29 g C m⁻² / mm) and deciduous needleleaf forest (DNF) has the least (mean SD = 0.15 g C m⁻² / mm) (Supplemental Figure S16).”

Figure 3: would be useful to have this map repeated for the remote sensing products in the supplementary. Do they show a more coherent spatial pattern compared to EC?

We have regridded the remote sensing data to a 0.5 degree spatial resolution and added this new Supplemental Figure S15 (below) to the supplemental section, along with some text:

L201: “Among the remote sensing products, there are not more visually consistent patterns of spring sensitivity of GPP to precipitation compared to the EC estimate or the TRENDY models (Supplemental Figure S15).”

Supplemental Figure S15: Maps of sensitivity of GPP in remote sensing products to spring precipitation. Note that MODIS MOD17 is MODIS-Terra.

L245: You could check from the data if the energy-limited sites that do not show increased productivity with spring drought are indeed particularly cold/cloudy sites.

Thank you for this helpful suggestion. We have followed your recommendation and now discuss both spring and summer in this section; we have included new supplemental figures S17 and S18 (see below).

L256-262: “Energy-limited ecosystems that increase GPP with precipitation during the spring have significantly lower PAR ($p = 0.018$) and air temperature ($p = 0.021$) than sites that decrease GPP with precipitation (i.e., increase GPP with meteorological drought; Supplemental Figure S17). In summer, energy-limited ecosystems that

increase GPP with meteorological drought have lower PAR ($p < 0.001$) but are not necessarily colder ($p = 0.22$) than ecosystems that do not (Supplemental Figure S18).”

Supplemental Figure S17: Energy-limited sites with positive GPP~precip sensitivities in the spring tend to have significantly lower PAR and air temperature than energy-limited sites with negative sensitivities in the spring. Comparing energy-limited site GPP~precip sensitivities, for sites with positive (>0) or negative (<0) sensitivities for (a) the sum of median spring PAR ($p = 0.018$) and (b) median spring air temperature ($p = 0.021$). Bars across the top indicate p-values, with both plots having significant differences at $\alpha = 0.05$.

Supplemental Figure S18: Energy-limited sites that maintain negative GPP~precip sensitivities during the summer receive less solar radiation ((a) lower PAR, $p < 0.001$), but (b) are not necessarily colder ($p = 0.22$). Points are individual EC sites and bars across the top indicate p-values.

L259: There are many papers for individual TRENDY models evaluating simulated GPP and ET against eddy covariance observations during water-stressed conditions. They generally point to land surface/ecosystem models responding too strongly to precipitation deficits (i.e. ET/GPP declining with rainfall, leading to a positive sensitivity) which may be one important factor explaining why the simulated sensitivities in energy-limited sites are generally more positive than observed at EC sites. This may imply a wrong sensitivity to precip declines rather than the models not capturing the co-benefits of precip decline (i.e. higher temp and/or radiation). This should be discussed as one possible explanation, making reference to past studies.

Thank you for this comment. We agree and have added additional discussion to place the results within the context of previous analyses. Please also see our response to this point raised in the reviewer's last main comment above.

L325: Should this be ERA-Interim?

Yes, this should be ERA-Interim, thank you for catching this (L378). This is what is used in the ONEFlux processing from Pastorello et al. (2020).

L328: what about cases where increasing GPP was driven by met conditions rather than plant growth (particularly at mature sites)? In this case you have detrended the GPP but not the corresponding met data. Can this introduce biases in the analysis?

Thank you for this thoughtful comment. We did not detrend the met data because of the relatively short time compared to climatological time scales (~60 years, Wu et al., 2007, <https://doi.org/10.1073/pnas.0701020104>) that are usually needed for detrending, and have added a new sentence clarifying this in the Methods (L383): “Meteorological data were not detrended because of the relatively short time series for climatology (Wu et al., 2007), and differences were minimal when testing detrending precipitation for spring (Supplemental Figure S19).”

However, to test whether there is any potential impact we performed the same analysis as the GPP detrending and detrended the precipitation data with a Mann-Kendall test and Sen’s slope.

Detrending precipitation during spring changes precipitation values and thus GPP sensitivities to precipitation for 4 sites: AT-Neu, BE-Vie, FI-Hyy, NL-Loo, but the resulting trend for a similar plot to Figure 1 for these sites is essentially the same as Figure 1 ($y = -1.26 * \log_{10}(x) - 0.08$; $R^2 = 0.47$, $p < 0.001$), so it doesn’t change anything in a meaningful way.

New regression for GPP~precip during spring: $y = -1.24 * \log_{10}(x) - 0.08$
 $R^2 = 0.48$, $p < 0.001$

Supplemental Figure S19: Detrending precipitation for the GPP~precipitation slopes in spring produces nearly an identical regression to Figure 1. The only sites that change their slopes are AT-Neu, BE-Vie, FI-Hyy, and NL-Loo.

L336: GPPsum is not defined in the text? For clarity, would it be better to use different “GPPsum” names for the P, PAR and Ta equations? E.g. Psum_P, Psum_PAR and Psum_Ta

It is the same GPPsum for each equation for each flux site. We now added the definition to the text, and also corrected air temperature in the equation ($T_{a\text{sum}}$ needed to be $T_{a\text{mean}}$).

L350: Do you mean the grid cell containing the site?

Yes, in nearly all cases, and we have revised the text to reflect this. There were some coastal sites (e.g., IT-Noe) that were just outside the gridding for TerraClimate and so we used the nearest grid cell in those few cases.

New L416: “WI was calculated using the grid cell containing each site (or nearest grid cell for coastal edges e.g., IT-Noe)...”

L370: please briefly state why SDGVM was excluded rather than just including the reference.

Thank you, this has been clarified by our co-author who processed the data. We now state (L441): “The model SDGVM was excluded because we observed an unrealistic drop in GPP after 2007.”

L372: Did you match the years to the EC sites or use 1992-2016 for all TRENDY site pixels? Are your results sensitive to this assumption if you didn't match the years?

We did not match years because some of the sites required going beyond 2016 to acquire 10 years of data, and TRENDY v6 ends in 2016. Results may be sensitive to the assumption that we did not match years, but we do not intend to predict the precise sensitivity at each individual site, we are more interested in the overall trend across the wetness index.

We added a new statement to make that clear for the readers (L445): “We did not match individual modeled years with the available years from the observed EC data; we focused on the overall trend in GPP sensitivities across the WI rather than the precise sensitivity at each EC site.”

Reviewer #2 (Remarks to the Author):

Miller et al. perform an analysis of spring GPP sensitivity to precipitation based on eddy-covariance (EC) data from >60 sites with >10 years observation distributed over Europe and North America and encompassing a variety of non-crop biomes. Precipitation sensitivity is defined as the slope of the linear relationship between spring GPP and spring precipitation. Miller et al. show that half of sensitivity variance across space can be predicted from a climate wetness index ($WI = P/PET$), with arid sites exhibiting a positive sensitivity, while wet sites exhibit a negative one. Miller et al. then compare EC sensitivity to that of a vegetation model ensemble and remote sensing and GPP products and show that models tend to overestimate the precipitation sensitivity of wet sites while remote sensing products underestimate that of arid sites. Finally, the authors use the fitted linear relationship between EC sensitivity and WI to extrapolate the sensitivity over the Northern Hemisphere and spatially characterize the mismatch between EC and model sensitivity.

While I found overall the paper to read well and to be interesting as it quantifies a perhaps under-studied phenomenon and provides useful benchmarking of vegetation models and remote sensing products, I have several concerns.

My first concern is regarding the definition of (meteorological) drought in the paper. While this is a key concept in the paper and is rightly so argued to matter for C cycling under climate change, drought is nowhere clearly defined in the paper. This is a (minor) issue in itself as meteorological drought can have several definitions and should thus be made clear from the beginning.

Thank you for bringing this important omission to our attention. We have revised the manuscript to include a clear definition, on L56:

“Theory suggests that GPP can increase during negative precipitation anomalies (i.e., meteorological droughts [(Wilhite and Glantz, 1985)]) in mid- and high-latitude ecosystems as well, a response that has found some support.”

A more in depth read of the paper suggests that drought is only indirectly defined through GPP sensitivity to precipitations. Thus, when the authors argue that GPP increases during spring drought in energy-limited sites, what they mean is that GPP is negatively correlated to precipitation at these sites. In my opinion this is not quite equivalent. Although this result does suggest that spring precipitation deficit enhances GPP, a more thorough definition of meteorological drought taking into account temperature and atmospheric water demand (e.g., based on meteorological drought indices such as SPEI or PDSI) might have yielded different results. This is even more relevant in the context of climate change, as globally drier conditions may arise essentially as a result of increased temperature and VPD, whereas changes in precipitation pattern will be more heterogeneous.

Thank you for this comment for improvement, we agree with the reviewer and have followed their suggestion to assess whether using a drought index influences the results. We thus downloaded the monthly PDSI product from TerraClimate for each of the sites and made an additional analysis (supplemental figures S7 & S8) substituting it for precipitation (following Figure 1). We used the mean PDSI for the spring (three months). Positive PDSI are wetter conditions, negative PDSI are drier (same direction as precipitation totals, see figures below). The trends we observe with GPP and precipitation hold when using PDSI, albeit with a weaker relationship.

Supplemental Figure S7: GPP sensitivity to TerraClimate PDSI during the spring. Following Figure 1, (a) sensitivity of each site along aridity gradient and (b) density of site sensitivities. 14/15 = 93% of the water-limited sites having positive GPP~PDSI relationships, and 33/46 = 72% of the energy-limited sites having negative GPP~PDSI relationships.

If we remove outliers by only taking the middle 95% of the sensitivities, we get a similar relationship:

Supplemental Figure S8: GPP sensitivity to TerraClimate PDSI during the spring, removing outliers by retaining only the middle 95% of site sensitivities. Following Figure 1, (a) sensitivity of each site along aridity gradient and (b) density of site sensitivities. 12/13 = 92% of the water-limited sites having positive GPP~PDSI relationships, and 31/44 = 70% of the energy-limited sites having negative GPP~PDSI relationships.

This analysis, using the 61 sites with 10+ years of data for spring, suggests that results we report with GPP and precipitation hold when using PDSI, albeit with a weaker relationship.

As shown by increasingly positive sensitivity of spring GPP to temperature and radiation across the climate wetness gradient, and as discussed by the authors, the paper main result is most likely driven by the positive effect of temperature and/or radiation at sites where water is not limiting. Taking into account the effect of temperature and radiation leads to much lower sensitivity to precipitation, in absolute value. Hence a more adequate conclusion would be that GPP at energy-limited sites is positively related to temperature, which is a well-known result.

Thank you and this interpretation is correct: higher temperatures and higher solar radiation co-occur with rainfall deficits (i.e., meteorological droughts). This is true as it is a well-known result that higher temperatures and PAR during spring lead to higher GPP, but we are approaching it from a meteorological drought perspective because the drought conditions themselves have been indicative of outlying increases in GPP in previous studies, without providing a clear mechanism behind them. Or in other words, drought can have beneficial effect on GPP depending on the overall limiting factors, which has not been shown in a systematic way across biomes yet. This is why we have set up our results this way.

We address this through the new PDSI plots (Supplemental Figure S7 and S8; Main text Lines 118-120) and new boxplots comparing spring and summer temperature and PAR for energy-limited sites (Supplemental Figure S17 and S18; Main text Lines 256-

262).

Comparison of EC vs model and remote sensing estimate of GPP sensitivity to precipitation is valuable. However, as is, I don't think that the spatial analysis of model-EC differences (i.e., Figure 3) adds much to the conclusion from Figure 2 that models tend to overestimate GPP sensitivity to precipitation. That is because vegetation models account for much more complexity than the EC extrapolation, which is based only on WI, does. Hence much of the spatial differences may be due to factors ignored by EC extrapolation. For example, from Figure 1, it seems that broadleaf forests tend to have a more negative sensitivity to precipitation than needleleaf, which may reflect contrasting species physiology or forest structure. Ignoring these differences may lead to overestimating the model error over needleleaf forests while underestimating it in the case of broadleaf. Soil factors (water capacity, rooting depth) are likely to be important factors as well of variations in GPP sensitivity to precipitation and driving model-EC differences.

We agree that there are many local factors (e.g., edaphic factors such as the mentioned water capacity and rooting depth) that we are not accounting for in our analysis with the EC extrapolation, and the TBMs allow for much more complexity in the sensitivity than the EC-WI regression. We feel however that our analysis still holds important value in that it allows for a direct comparison to remote sensing and TBM-derived values; this can be used as a point of comparison through which the influence of other local factors can be further examined in future work. There are likely differences in plant function, structure, and diversity of species that we do not address, but the scope of these differences is beyond our current analysis and we feel would be best reserved for a future study. We address this will a new paragraph in the Discussion, L317:

“We upscaled estimates of spring GPP sensitivity to precipitation from EC and the WI to provide a first-order approximation of spatial patterns across the Northern Hemisphere (Figure 3). These estimates can be directly compared with results from TBMs and remote sensing and can be used as a basis for future comparisons (e.g., Wolf et al., 2016; Bastos et al., 2020^{30,35}). However, we acknowledge that, beyond aridity, there are many local environmental factors that vary across different regions and are incorporated by TBMs and remote sensing observations—this includes edaphic or microclimatic factors (e.g., soil texture, water availability, plant structure and diversity) (Fisher et al., 2014; Stocker et al., 2019; Piao et al., 2013; Lian et al., 2021)^{37,48,53,60}. Within individual TBMs, the diversity of these responses will depend on how variables are parameterized (Fisher et al., 2014)³⁷. Our results can be used as a reference to evaluate what would be expected from a climatological perspective (i.e., water vs. energy limitation) alone, and there remains unexplained variability in our regression (Figure 1).”

We have added new supplemental figures comparing the SD of sensitivities spatially (Supplemental Figure S14) and across biome types in TRENDY models (Supplemental

Figure S16), and we also have maps for remote sensing data products (Supplemental Figure S15).

Reviewer #3 (Remarks to the Author):

Summary:

This study integrates GPP datasets from in-situ EC flux towers, remote sensing, and DGVM simulations to investigate the sensitivity of spring GPP responses to meteorological drought. The flux tower data analysis shows that the 'gross' sensitivity of spring (MAM) GPP to inter-annual variability in precipitation, calculated with simple linear regression, is highly correlated with site aridity (wetness index). In moist, more energy-limited sites, the GPP~precipitation sensitivity can be negative, which is mostly driven by associated increases in temperature during such meteorological drought. Comparison with TRENDY model simulations suggests the models overall underestimate the magnitude of such negative precipitation sensitivity. Meanwhile, the remote sensing data set can capture the negative sensitivity but cannot capture the positive sensitivity in water-limited ecosystems. Finally, the study analyzes spatial patterns of model-data differences in spring GPP sensitivity to precipitation by empirically scaling site-level sensitivity to the whole northern hemisphere.

Comments:

The paper is well written. The analysis is comprehensive and careful given so many different data sets are involved. The figures are nice and easy to read.

My main concern is that the lack of unwrapping of 'drought' in the title and main text is misleading and confusing. For example, Fig. 1&2 both focus on the idea that models cannot capture the 'negative' GPP sensitivity to precipitation. However, as the authors recognized in the study, 'meteorological drought' is often accompanied by higher radiation and temperature. In fact, Fig. S7 nicely shows the partial effect of GPP to water, radiation, and temperature. It is interesting to see that the average partial sensitivity of GPP to precipitation in energy-limited systems is not negative but around 0 (Fig. S7 panel) while TRENDY model is positive. This indicates that the models are too sensitive to moisture and underestimate sensitivity to temperature (e.g. phenology). I am worried that the current framing of the manuscript can result in an erroneous interpretation that 'moisture-dependence' is wrong in the models, which is not the case based on Fig. S7. Overall, I suggest using the 'partial sensitivities' instead of gross sensitivities in the main text.

Thank you for this comment, and we acknowledge the significance of both gross sensitivities and partial sensitivities in this study. While our paper primarily emphasizes the sensitivity of precipitation in relation to meteorological drought and GPP, we also address the influences of temperature and PAR to encompass partial sensitivities. Furthermore, we have incorporated a PDSI regression (Supplemental Figure S19) to provide a more technically defined assessment of drought.

We find that the gross (univariate) sensitivity moisture dependence in the TBMs for GPP ~ precipitation in the spring is not sufficiently negative for energy-limited ecosystems compared to eddy covariance (and remote sensing; Fig. 2). With the partial (multivariate) sensitivity, the TBMs always have positive GPP~precip sensitivities for energy-limited sites, while energy-limited EC sites are centered around zero (Supplemental Figure old S7, S9 in new numbering). We agree that the TBM GPP is too sensitive to moisture in the partial sensitivities, but it is not sensitive enough in the gross (univariate) sensitivities. Based on Figure old S7 (new S9), the moisture dependence is too positive in models, when it should be more neutral; based on Figure 2, the moisture dependence is not negative enough in models; this is the same direction (shifted more positive) in both cases.

We also agree that the models are not sensitive enough to temperature compared to EC, but this is not the primary focus of our paper and many studies have examined model sensitivities to temperature (e.g., Buermann, W., Forkel, M., O'Sullivan, M. *et al.* Widespread seasonal compensation effects of spring warming on northern plant productivity. *Nature* **562**, 110–114 (2018). <https://doi.org/10.1038/s41586-018-0555-7>). In the revised manuscript, we further emphasize the importance of the partial sensitivities in the Results and Discussion (e.g., Lines 121-134, 248-268), and discuss how our use of gross sensitivity to precipitation should be interpreted in the relative context of partial sensitivities, including other environmental factors (e.g., VPD, energy balance).

Some minor comments:

Line 29. this is inaccurate, drought does not increase GPP. Associated temperature and radiation increase (mostly temperature based on the analysis) drives GPP increase.

Thank you, we have rephrased this to: “GPP increases during spring drought, primarily due to the associated increases in temperature and radiation during periods of anomalously low precipitation, ...”

Line. 43-44. To me, energy-limited denotes radiation-limited (e.g. the classic paper by Nemani et al. 2003) in contrast to temperature and moisture limited. Temperature is surely one kind of energy limitation. It is ok for such a generalization but more clear definition would be better here.

Thank you for this suggestion. There are various factors to include in the limitations of energy, and Nemani et al. (2003) is one perspective we have cited here, as well as Churkina and Running (1998) and the more recent Donohue et al. (2013) and Sippel et al. (2018). We have added “(e.g., radiation, temperature)” after energy limitations in the sentence to be more explicit. Both radiation and temperature are included in the Budyko framework for aridity due to PET scaling, which is how we are deriving our definitions here (see Sippel et al. 2018 <https://doi.org/10.1007/s40641-018-0103-4> for more details).

Line 91, Fig.1 Why not put 4 rows in this figure? First row is gross sensitivity to precipitation, then followed by partial sensitivity to precipitation, radiation, and temperature. As I expressed above, this unwrapping is important.

Thank you for this suggestion. To the left is a mock-up of what Figure 1 might look like if we included the partial sensitivities. While this does show more of the details regarding the differences between precipitation, temperature, and PAR in how they combined to affect spring GPP, with respect we feel that it would take the focus off the main message of the paper, which is the gross precipitation response (in the top panel).

We agree that unwrapping these different sensitivities is an important point to characterize the results, and this is why we have included details related to Figure old S7 (new S9) in the supplemental material immediately after presenting this result. The temperature sensitivity itself would not be a new result in the previous literature, but the consistency of the precipitation response across EC sites (our main takeaway) is new because we are able to show it with observational data.

The partial sensitivities could also be argued to be incomplete because they do not account for all differences between ecosystems, or compensating effects, and there are other factors that differ between sites that would not be accounted for by the partial relationships of precipitation, temperature, and PAR (e.g., VPD, edaphic factors). A complete analysis of this form would require a detailed modeling or machine learning approach which we feel is beyond the scope of the present study.

Line 106-107, 'Few energy-limited', there is still a quarter of sites, which is more than 'few'.

Changed to “relatively few”

Line 157, I am curious why some models have closer sensitivity to EC? e.g. JSBACH. Does JSBACH have better moisture dependence and phenology? Include such a figure for partial sensitivity would be helpful as well.

We now added discussion of TRENDY models in the Discussion, and including this sentence on JSBACH, L299: “JSBACH in particular does well in comparison to the EC data; this may be due extensive parameterization for individual plant functional types (PFTs) and different stomatal conductance calculations under stressed and unstressed soil water conditions; however, the specific attribution is beyond the scope of this analysis and we refer to Reick et al.⁴⁸ (2021) for more details on this model.”

Line 330-331. Why meteorological data was not detrended?

Thank you for this comment, and Reviewer 2 had a similar comment and we include our response below.

We did not detrend the met data because of the relatively short time compared to climatological time scales (~60 years, Wu et al., 2007, <https://doi.org/10.1073/pnas.0701020104>) that are usually needed for detrending, and have added a new sentence clarifying this in the Methods (L383): “Meteorological data were not detrended because of the relatively short time series for climatology (Wu et al., 2007), and differences were minimal when testing detrending precipitation for spring (Supplemental Figure S19).”

However, to test whether there is any potential impact we performed the same analysis as the GPP detrending and detrended the precipitation data with a Mann-Kendall test and Sen’s slope.

Detrending precipitation during spring changes precipitation values and thus GPP sensitivities to precipitation for 4 sites: AT-Neu, BE-Vie, FI-Hyy, NL-Loo, but the resulting trend for a similar plot to Figure 1 for these sites is essentially the same as Figure 1 ($y = -1.26 * \log_{10}(x) - 0.08$; $R^2 = 0.47$, $p < 0.001$), so it doesn’t change anything in a meaningful way.

New regression for GPP~precip during spring: $y = -1.24 * \log_{10}(x) - 0.08$
 $R^2 = 0.48$, $p < 0.001$

Supplemental Figure S19: Detrending precipitation for the GPP~precipitation slopes in spring produces nearly an identical regression to Figure 1. The only sites that change their slopes are AT-Neu, BE-Vie, FI-Hyy, and NL-Loo.

REVIEWER COMMENTS

Reviewer #1 (Remarks to the Author):

I have no further comments on this manuscript and recommend it for publication

Reviewer #2 (Remarks to the Author):

Overall, the authors have addressed all comments in a satisfactory manner. The manuscript clarity has improved and the discussion and conclusions are robust to the suggested changes in the analysis. I still only have a couple minor comments. Thus, I recommend this manuscript for publication in Nature Communications with minor revisions.

I agree that spatial comparison of EC and model GPP sensitivity to precipitation holds value despite its limitations. The added paragraph makes a good job at highlighting these limitations. However, please consider the following slightly different approach, which may be more satisfying. Instead of calculating a map of model vs EC deviations based on the difference between interpolated EC sensitivity and model sensitivity, why not try to first regress deviations against WI and then interpolate based on this regression? I think this would provide the advantage of more accurately represent the expected effect of WI on model-EC deviations across biomes.

The result mentioned L131-134 that spring GPP still decreases with precipitation, even when accounting for temperature and radiation is interesting but I think that discussion of potential mechanisms is missing in the Discussion. This could pinpoint to what process is lacking in TBMs that lead to TBMs not reproducing this negative sensitivity to precipitation as mentioned L184.

Reviewer #3 (Remarks to the Author):

The revision improves the manuscript, especially with added information on drought definition and model differences. Yet, as I expressed in my comments before, I still think focusing mainly on 'gross' sensitivity undermines the comprehensive analysis and can be misleading.

For instance, the current abstract ends with "our results show that GPP in TBMs may not be sufficiently sensitive to spring droughts". "not sufficiently sensitive to spring droughts" is very vague. Spring drought is a syndrome incorporating various changes in climate forcing (temperature, precipitation, etc.). So, is the model not sensitive enough to water or temperature? I think the authors have the answer in the partial sensitivity analysis, which should be highlighted more.

In addition, the result section (Line 132 - 134) says "Therefore, our results show that although positive temperature anomalies are the main control on GPP sensitivity during spring for energy-limited sites, meteorological drought is an important and thus far underestimated secondary control for ecosystem productivity, particularly as it often co-occurs with warming." Please show evidence that meteorological drought is a far underestimated secondary control.

Finally, I suggest using "meteorological drought" instead of drought in the title. Again, I am just worried that the manuscript leads to a misinterpretation that erroneous moisture dependence in TBMs led to biases in spring drought

REVIEWER COMMENTS

Reviewer #1 (Remarks to the Author):

I have no further comments on this manuscript and recommend it for publication

Thank you again for your earlier comments on our manuscript and for your recommendation for publication.

Reviewer #2 (Remarks to the Author):

Overall, the authors have addressed all comments in a satisfactory manner. The manuscript clarity has improved and the discussion and conclusions are robust to the suggested changes in the analysis. I still only have a couple minor comments. Thus, I recommend this manuscript for publication in Nature Communications with minor revisions.

I agree that spatial comparison of EC and model GPP sensitivity to precipitation holds value despite its limitations. The added paragraph makes a good job at highlighting these limitations. However, please consider the following slightly different approach, which may be more satisfying. Instead of calculating a map of model vs EC deviations based on the difference between interpolated EC sensitivity and model sensitivity, why not try to first regress deviations against WI and then interpolate based on this regression? I think this would provide the advantage of more accurately represent the expected effect of WI on model-EC deviations across biomes.

Thank you for your suggestion. Although we understand the motivation for your suggestion, we feel that there are some important aspects of the suggested analysis that undermine its utility with respect to identifying the expected effect of WI on model-EC deviations across biomes.

In our current approach, we generate a map of expected spring GPP~precipitation sensitivity from the linear regression of this sensitivity to the WI, derived from EC observations, by extrapolating across a grid of WI values from TerraClimate (Figure 3a). We then subtract this from the TRENDY model mean GPP~precipitation sensitivity (Figure 3b) to yield a difference map (Figure 3c). Importantly, this method retains the spatial variability that is present in the TRENDY model gridded data at 0.5 degree spatial resolution - essential information that would be lost in the suggested method.

Using the method suggested, we would first calculate the difference between TRENDY model mean and EC-estimated GPP~precipitation sensitivities for each EC site. While this would give a difference of the estimates for each site, the resulting value would incorporate error from both TRENDY *and* from EC sensitivities. In our current approach, we do not make inference from the sensitivities for each individual site, but instead our results are driven by the trend across the aridity gradient (i.e., WI). With the suggested approach, we expect to have more error because of the very different spatial scales between the EC sites themselves (hundreds of meters) and the TRENDY estimates (0.5 degree, or roughly 25 km). While a comparison in the overall trends across the WI can be made, a comparison of the GPP~precipitation sensitivity from TRENDY and from EC is not appropriate for each individual site because they are measuring very different areas. A different sensitivity should perhaps thus be expected, and neither the model nor the EC data may be incorrect despite their differences. Extrapolating such data spatially would thus be quite questionable.

We have however implemented the approach suggested by the reviewer to assess how the analysis adds to the interpretation of the results. If we subtract the EC GPP~precip slopes from the TRENDY model mean GPP~precip slopes as suggested, and do a linear regression with the log10 of the wetness index (WI), this yields the equation:

y = difference in slopes, TRENDY - EC

x = WI

y = 0.35*log10(x) + 0.15, R2 = 0.058, p = 0.061

Applying this to the TerraClimate grid of WI, following the reviewer's suggestion, yields the following map (with the same ranges as our original approach presented in Figure 3c):

Top is the new map based on the reviewer's suggestion, and bottom is the difference map (original version of Figure 3c). The color scales are the same in both figures.

As we hypothesized, much of the spatial information is lost and the resulting map is primarily dominated by spatial changes in WI. We would prefer not to include the new suggested map because it does not adequately describe spatial differences between the EC-estimated slopes and the TRENDY model mean slopes.

The result mentioned L131-134 that spring GPP still decreases with precipitation, even when accounting for temperature and radiation is interesting but I think that discussion of potential mechanisms is missing in the Discussion. This could pinpoint to what process is lacking in TBMs that lead to TBMs not reproducing this negative sensitivity to precipitation as mentioned L184.

Thank you for this comment. In the Discussion section, starting at Line 270 (in the updated clean draft), we emphasize how the sensitivities observed in TBMs differ from the EC data, and why this might be in the case. In the following paragraph, starting at Line 288, we discuss several possible reasons why different models are not reproducing the negative sensitivity to precipitation we observe (mentioned in Line 184, now Lines 182-187). For example, we mention that LPX has a positive correlation between photosynthesis and water availability by design in this model. But this is not necessarily the case for other models because of differences in design, and we mention a few cases to try to attribute the responses we observe. Each model likely has a subtly

different reason as to why it does not exactly emulate the processes observed by the EC, as this form of attribution for each model is beyond the scope of our analysis since this is an observational comparison study rather than a model development study.

Paragraph at Line 270:

TBMs are a key tool in predicting the impacts of climatic change on the carbon cycle^{37,38,41}. We found that TBMs from the widely used TRENDY ensemble consistently underestimated the strength of spring GPP sensitivity to precipitation deficits in energy-limited ecosystems observed by EC. Despite having much larger grid cells (0.5°) than the flux tower footprints⁴⁶, the TBMs had an accurate span of positive GPP sensitivities for water-limited sites, but not negative sensitivities for energy-limited sites. This suggests that GPP-precipitation relationships in many TBMs may be discounting the potential co-benefits of precipitation reductions on spring GPP, and solely focus on precipitation as increasing GPP, which is well-represented for water-limited ecosystems. As a consequence, TBMs may also underestimate the risks of spring drought contributing to later loss of productivity during the growing season, such as through structural overshoot^{45,47}. A possible explanation is that TBMs may reduce GPP too strongly under precipitation deficits, which might suggest an excessive GPP sensitivity to precipitation reduction rather than due to co-occurring increases in temperature and solar radiation; for example, in their meta-analysis, Piao et al.⁴⁸ found that all 10 of the studied TBMs had a positive GPP sensitivity to variations in precipitation in the northern hemisphere and that TBMs were more sensitive to precipitation than temperature. In addition, the regression of net biome productivity, which is influenced by GPP, to precipitation was overestimated in 9 of 10 of their⁴⁸ studied models compared to the observed residual land sink from Friedlingstein et al.⁴⁹.

Paragraph at Line 288:

We find a wide array of GPP-precipitation sensitivities across models, with several outliers in how individual models respond to spring drought (**Figure 2; Supplemental Figure S8**). Under conditions of water stress, LPX downregulates photosynthesis to match water demand and supply since it has a supply and demand driven approach⁵⁰. Therefore, photosynthesis and water availability have a positive correlation by design in this model, which may be contributing to the positive correlation between spring GPP and precipitation we observe in **Figure 2** for both energy- and water-limited ecosystems. LPJ, LPJ-GUESS, and ORCHIDEE modify the maximum rate of carboxylation (V_{cmax}) to environmental conditions for photosynthetic carbon gain^{37,50}, and this likely influences the higher sensitivity of GPP to changes in precipitation in these models in water-limited regions. ORCHIDEE-MICT modifies V_{cmax} as well⁵⁰, but it does not have the high sensitivities for water-limited sites that are present in ORCHIDEE; this may be due to the higher water holding capacity of soils in

ORCHIDEE-MICT⁵¹, such that water-limited sites are more buffered against precipitation deficits. JSBACH in particular does well in comparison to the EC data; this may be due to extensive parameterization for individual PFTs and different stomatal conductance calculations under stressed and unstressed soil water conditions; however, specific attribution is beyond the scope of this analysis and we refer to Reick et al.⁵² for more details on this model. Other models that adjust for stomatal conductance (e.g., CABLE, DLEM) also do well for water-limited sites⁵⁰, but do not have the sensitivity for energy-limited sites seen in JSBACH.

Lines 182-187:

Even when accounting for PAR and air temperature, sites in the EC data can have negative spring GPP sensitivities to precipitation, while this is never the case in the TRENDY model ensemble mean (Supplemental Figure S9). In other words, the models indicate that more precipitation would uniformly correspond to more GPP during spring across the entirety of our aridity gradient, while this is not observed at EC sites.

Reviewer #3 (Remarks to the Author):

The revision improves the manuscript, especially with added information on drought definition and model differences. Yet, as I expressed in my comments before, I still think focusing mainly on 'gross' sensitivity undermines the comprehensive analysis and can be misleading.

For instance, the current abstract ends with "our results show that GPP in TBMs may not be sufficiently sensitive to spring droughts". "not sufficiently sensitive to spring droughts" is very vague. Spring drought is a syndrome incorporating various changes in climate forcing (temperature, precipitation, etc.). So, is the model not sensitive enough to water or temperature? I think the authors have the answer in the partial sensitivity analysis, which should be highlighted more.

We have edited the Abstract to clarify that we are specifically looking at precipitation deficits. The associated sentence now reads (L27-30): "While remote sensing products represent the negative spring GPP sensitivity for energy-limited ecosystems well, our results show that GPP in TBMs may not be sufficiently sensitive to precipitation deficits during spring."

We discuss the partial sensitivities in the Results and Discussion and do not highlight them to the full extent in the Abstract, largely because they are supportive of the main takeaways and the brevity of the Abstract necessitates that they be left out. The models

are not sensitive enough to precipitation and they are quite sensitive to temperature.

In addition, the result section (Line 132 - 134) says "Therefore, our results show that although positive temperature anomalies are the main control on GPP sensitivity during spring for energy-limited sites, meteorological drought is an important and thus far underestimated secondary control for ecosystem productivity, particularly as it often co-occurs with warming." Please show evidence that meteorological drought is a far underestimated secondary control.

We feel there has been a misunderstanding here, related to phrasing in this sentence, and we have inserted commas to better emphasize the intended meaning. The sentence intended to convey that "meteorological drought is an important and, thus far, underestimated secondary control" (meaning: to date, it is underestimated) instead of meteorological drought being far underestimated as a secondary control. While it is underestimated, we agree that it is not "far" (i.e., greatly) underestimated as a secondary control.

Our evidence for meteorological drought as underestimated secondary control is highlighted in the sentences prior to the one stated in the comment:
"Even when controlling for PAR and air temperature in a multiple linear regression in the spring EC observations (see Methods), 61% of energy-limited sites ($n = 28/46$) have a negative GPP sensitivity to precipitation (Supplemental Figure S9). By contrast, in the partial spring GPP sensitivity to air temperature, 93% ($n = 43/46$) of energy-limited sites have a positive sensitivity. For energy-limited sites, this suggests that warmer springs almost always increase GPP when controlling for precipitation (93%) compared to drier springs when controlling for air temperature (61%)."

Finally, I suggest using "meteorological drought" instead of drought in the title. Again, I am just worried that the manuscript leads to a misinterpretation that erroneous moisture dependence in TBMs led to biases in spring drought

We appreciate this suggestion and understand the Reviewer's desire for clarity in the title of our manuscript. However, with respect, we wish to keep the title as it currently is and not insert "meteorological" in the title. We have three main reasons for this: (1) there are many different definitions of drought in various disciplines (e.g., ecosystem science, agriculture, hydrology, meteorology), and we wish to keep the paper as widely applicable to many audiences and not artificially restrict to those only interested in meteorological drought, which itself can also have variable definitions in the literature; (2) we clearly state that we are discussing "meteorological drought" in both of the first two sentences of the abstract, and so the form of drought we are investigating should

be apparent to those who read beyond the title; and (3), our opinion is that inserting “meteorological” into the title would make it many syllables longer and more unwieldy than it currently is.